# Presence and Quantification of Microplastic in Urban Tap Water: A Pre-Screening in Brasilia, Brazil

**Claudia B. Pratesi** [1,2,*], **Maria Aparecida A. L. Santos Almeida** [1,2], **Geysa S. Cutrim Paz** [1,2], **Marcelo H. Ramos Teotonio** [1,2], **Lenora Gandolfi** [1,2,3], **Riccardo Pratesi** [1,2,3], **Mariana Hecht** [1,3,*] **and Renata Puppin Zandonadi** [4,*]

1   Interdisciplinary Laboratory of Biosciences, School of Medicine, Campus Universitário Darcy Ribeiro, University of Brasilia, Asa Norte, Brasilia 70910-900, DF, Brazil; cidaleite08@gmail.com (M.A.A.L.S.A.); geysa94@hotmail.com (G.S.C.P.); Marceloswiss@gmail.com (M.H.R.T.); lenoragandolfi1@gmail.com (L.G.); pratesiunb@gmail.com (R.P.)
2   Post-Graduate Program in Health Sciences, School of Health Sciences, University of Brasilia, Brasilia 70910-900, DF, Brazil
3   Post-Graduate Program in Medical Sciences, School of Medicine, University of Brasilia, Brasilia 70910-900, DF, Brazil
4   Department of Nutrition, Faculty of Health Sciences, Campus Darcy Ribeiro, University of Brasilia (UnB), Asa Norte, Brasilia 70910-900, DF, Brazil
*   Correspondence: cpprates@eckerd.edu (C.B.P.); marianahecht@gmail.com (M.H.); renatapz@unb.br (R.P.Z.); Tel.: +1-407-230-1661 (C.B.P.)

**Abstract:** Plastic pollution is a rapidly growing environmental and human health crisis, with no sign of improvement. From 2012 to 2020, the number of studies on plastic pollution increased, and macro to nano-sized plastics have been documented in the most remote biomes of the planet. Studies have shown contamination by microplastics (MPs) in various types of food consumed by humans, including seafood, honey, sugar, salt, tap and bottled water and beer. This study's objective was to detect the possible contamination by MPs in drinking water samples collected from two main residential and commercial areas of Brasilia. A total of 32 samples (500 mL) of tap water were collected from residential and commercial areas. Samples were processed and transferred to a Sedgewick-Rafter counting cell chamber. The presence of MP particles was analyzed using a Nikon Eclipse fluorescence microscope. MPs were found in 100% of the samples. The mean microplastic particles per 500 mL found in the South Wing area was $97 \pm 55$, while the mean number of particles in the North Wing area was $219 \pm 158$, and the MPs found ranged in size from 6–50 microns. The study results reveal a disturbing amount of MP particles in Brasilia's tap water. This surprising number of particles in residential and commercial tap water is especially considering that tap water is not the only source of MPs to which people are exposed.

**Keywords:** microplastic; tap water; plastic pollution; Brazil

## 1. Introduction

Plastic pollution is a rapidly growing environmental and human health crisis, with no sign of improvement [1–4]. According to He et al. (2020), from 2012 to 2020, the number of studies on plastic pollution increased from 19 to 1221 per year [5,6], and plastic pollution, from macro to nano size (macroplastics: 5–50 cm; mesoplastics: 0.5–5 cm; microplastics: 1 μm–5 mm; and nanoplastics: <1 μm) [7], has been documented even in the most remote biomes of the planet [8–11]. Studies have shown contamination by microplastics (MPs) in mollusks [12], crustaceans and fish used in human nutrition [13,14] and also in tap water, bottled water [15], beer [16], honey, sugar [16] and salt [17,18]. Plastic material is cheap, lightweight, durable and resistant to corrosion, breakage and biodegradation [15]; therefore, it never goes away. When exposed to the sun and wind, plastic undergoes photo-oxidation,

where progressively, larger pieces of plastic break down into smaller and smaller micro- or nano-particles [19,20].

MP effects on the environment and organisms have not yet been sufficiently studied. However, studies have shown that MPs adsorb harmful substances that may accumulate in fatty tissue and cause harm to organisms [21,22]. Additionally, additives in plastics can have toxic effects on humans [23,24], and ingestion of MPs could cause mutagenic effects [25,26]. They can also provide a long-term stable habitat for various harmful microorganisms and pathogens [27,28].

Nevertheless, because of its versatile properties, consumers have become dependent on plastic. World production had gone from a million tons when production began in 1945 to an alarming rate of 335 million tons in 2017 [29]. With the increase in production, there was also an increase in plastic waste [2,30–32]. It is estimated that more than 10% of the world's garbage is made of plastic [30], with approximately 120 million tons of post-consumer plastic waste [33].

This exponential growth in the production and consumption of plastics has created several problems. MP debris comprises non-renewable source material containing additives from different chemicals that travel great distances and are released and accumulate in natural environments [30]. The occurrence of plastic debris has been reported in several aquatic environments, including rivers, lakes, estuaries and coastal and marine ecosystems, and represents an increasing environmental concern [2,4,30,34].

Plastic pollution is not new, and for over five decades, environmentalists have been sounding the alarm about its potential environmental impact, and more recently, its health impact. However, due to its multidisciplinary nature, there are still several knowledge gaps [32] that need to be filled. Tap water contamination is of great concern because of its daily consumption, including its uses in cooking and washing fruits and vegetables. Despite evidence of widespread contamination [18,35–38], we are still unsure about the health effects of this steady, daily MP diet [39]. However, Swan and Colino (2021), in their book *Count Down*, argue that continuous exposure to endocrine disrupters such as Bisphenol A and phthalates found in plastic is responsible for, among others, increased infertility problems and obesity rates [40]. Studies show that annual human consumption of microplastics, depending on age and sex, ranges from 39,000 to 52,000 particles [41–44].

Brasilia, inaugurated in 1960, is the capital of Brazil and the seat of the federal government. This planned city is shaped like an aircraft [45]. The body of the airplane is composed of a central administrative area where downtown is located, while the federal government is located in the 'cockpit'. The residential and commercial sectors on each side of the central area are known as Asa Sul (South Wing) and Asa Norte (North Wing). Middle- and upper-middle-class residents inhabit Brasilia's wings, which are presently home to approximately 220 thousand people. This initially planned nucleus suffered a progressively accelerated and mostly disorganized process of peripheral urbanization, and currently, the greater Brasilia population has reached close to three million. During the last few decades, the progressive urbanization around the city's water basin has increased the pollution of its streams and reservoirs and has caused a severe water supply crisis. Despite coming from the same water system treatment, the South Wing was built earlier than the North Wing, so the plumbing systems are different, and water is frequently stored in water tanks (made of stainless steel, polyethylene, reinforced polyester, fiber cement and fiberglass). Therefore, we hypothesize that the microplastic content could be higher in the North Wing than in the South Wing. Moreover, in the same region it is possible to have different concentrations of microplastics for the final consumer.

Our study's objective was to quantify and investigate the presence of microplastic in urban tap water samples collected from two main residential and commercial areas of Brasilia.

## 2. Materials and Methods

The South and North Wings, located on each side of the city's central area, are part of the city's initial nucleus and are composed of 60 superblocks per wing (about 13.5 km of extension), with an average of 10 apartment buildings in each. Superblocks are surrounded by small commercial areas formed by local stores, restaurants, bars, cafes and bakeries. Tap water samples were randomly collected (not with a pre-determined collection algorithm) from bars, restaurants and cafes in the commercial areas adjacent to the superblocks. Bars, restaurants and cafes were chosen as collection points due to the convenience of obtaining samples compared to residential sites. In North Wing and South Wing, water is captured from water sources and transported through pipelines to the water treatment station (in this region by flotation and direct filtration) to be later stored and distributed to consumption areas [46].

The South Wing was built earlier than the North Wing [47], so the plumbing systems are different. Additionally, water is frequently stored in water tanks (made of stainless steel, polyethylene, reinforced polyester, fiber cement and fiberglass).

A total of 32 samples of tap water were collected, 16 samples in the South Wing during the second half of January 2018, and 16 samples in the North Wing during the first half of February 2018. Although the two areas received water from the same water treatment plant, the samples were taken on different days. Collection sites can be seen in Figure 1, marked by the red dots.

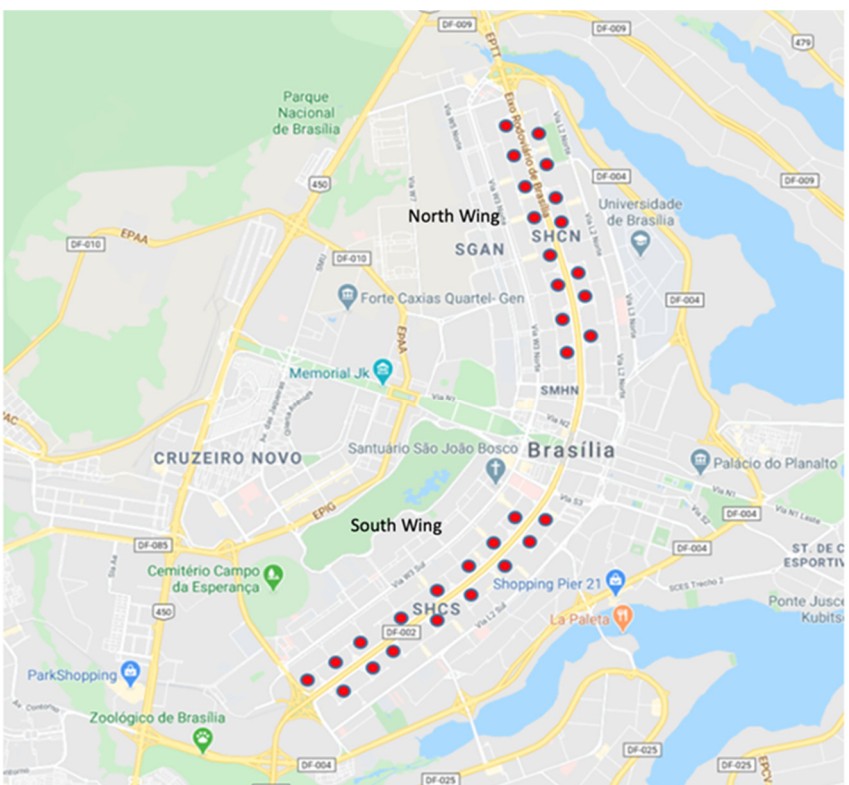

**Figure 1.** Airplane-like design of Brasilia showing the sites of tap water collection in the South and North Wings of the main residential area. All samples were randomly collected from local bars, restaurants and cafes.

The potable water of the area under study comes from the same water treatment plant (Water Treatment Unit No. 1) belonging to the Environmental Sanitation Company of the Federal District (CAESB). All the company water treatment units (a total of 10) use the same direct filtration system. This treatment consists of the addition of a coagulant, rapid mix, flocculation, filtration and fluoridation as a final step before distribution.

### 2.1. Sanitation and Contamination Prevention

The possibility of organic contamination was considered remote, given the precautions set in place. To ensure there was no contamination, negative blanks with Milli-Q water were randomly mixed in while samples were processed. Furthermore, during collection, preparation and analysis of the samples, all contact and contamination with plastic materials were avoided. Before the collection of the samples, all bottles were washed and rinsed several times with Milli-Q water. Instruments and work surfaces were cleaned with 100% alcohol. Lab coats (cotton material) and latex gloves were worn during the study. All bottles, instruments and surfaces were cleaned with Milli-Q water after each sample was analyzed to avoid contamination. A laminar flow hood was used during the analysis process (Pachane Biotechnology Pa410, Piracicaba, SP, Brazil). Ten samples of Milli-Q water were prepared to ensure no contamination happened during sample collection, preparation, or analysis. These samples were analyzed using the same procedure as for tap water samples, including storage in the sample collection bottles. All samples, including the Milli-Q water negative controls, were randomly identified with a sticker and a code to avoid the examiner's possible bias. Duplicate samples were collected in 500 mL laboratory amber bottles. The bottles were filled with tap water for 1 min until they overflowed. With the water still running, the bottle was filled and dumped twice before being filled a third time and immediately capped with a glass cap.

### 2.2. Sample Processing

Water from each bottle was transferred inside the laminar flow hood to a 500 mL flat bottom flask, adding 1.37 g mL$^{-1}$ of ZnCl$_2$ [48]. The mixture was shaken vigorously and allowed to settle overnight.

After that, 800 µL of the supernatant was carefully collected and added to 200 µL of Nile red dye (technical grade, N3013, Sigma Aldrich) in methanol, with a concentration of 10 µg/mL$^{-1}$ [49]. This final solution was transferred into a 5 mL Eppendorf tube, briefly vortexed and left to stand for 10 min. The solution was transferred to a Sedgewick-Rafter counting cell chamber. The Sedgewick-Rafter chamber has 1000 quadrants with a total volume of 1 mL, and each quadrant corresponds to 1 µL. The presence of MP particles was analyzed with the Nikon Eclipse Ni-U fluorescence microscope using the FITC filter (wavelength: 475–625 nm).

The visualization was made with a magnification of 20–200×. One hundred random quadrants, corresponding to 10% of the total area of the chamber, were counted. A simple percentage calculation was made to determine the final number of MP particles in each 500 mL sample.

### 3. Results

All 32 samples (100%) analyzed were positive for the presence of MPs. The mean MPs per 500 mL found in the South Wing area was 97, while the mean number of particles in the North Wing area was 219 (Table 1). The MPs found ranged in size from 6–50 microns; however, some MPs were extremely small, and therefore it was not possible to determine their size or quantity (Figure 2). Additionally, due to the method used, it is estimated that the heavier particles were not captured; consequently, the number of particles was likely underestimated [50,51].

**Table 1.** Mean, median and SD values of MP particles found in 500 mL samples of tap water in the city of Brasilia, Brazil.

|  | **Mean** | **Median** | **Max** | **Min** | **Standard Deviation** |
|---|---|---|---|---|---|
| South Wing | 97 | 78 | 228 | 24 | ±55 |
| North Wing | 219 | 161 | 597 | 48 | ±158 |

Figure 2 shows microplastic particles fluorescently dyed using Nile Red. Images were taken at 20–200 magnification.

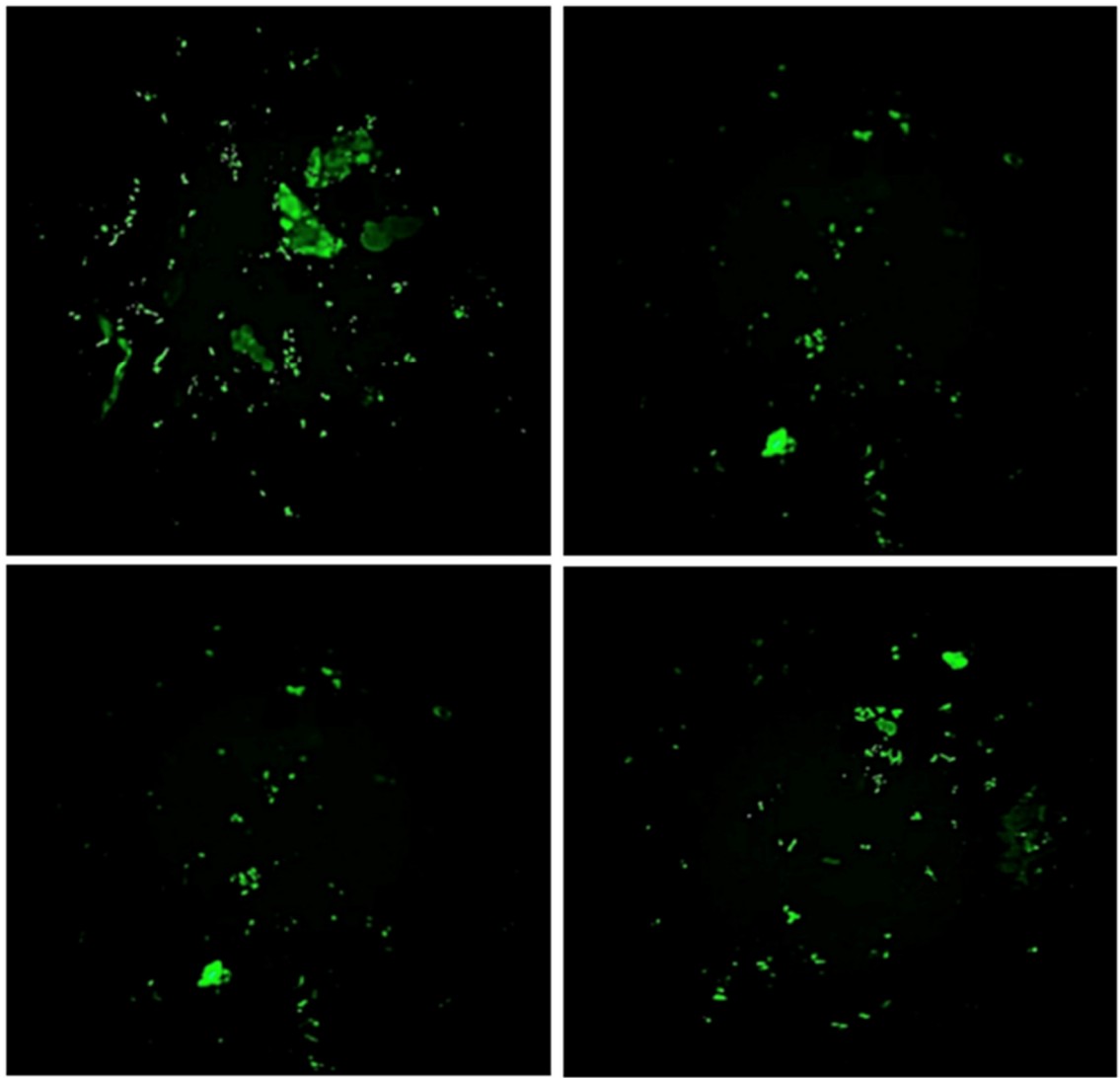

**Figure 2.** Fluorescence-dyed microplastic particles found in tap water samples in the city of Brasilia, Brazil.

### 4. Discussion

Our results showed that all samples presented microplastic particles; however, the water from the North Wing area presented a higher mean of MPs (219 MP/500 mL of water) than the South Wing (97 MP/500 mL of water), and MP size ranged from 6–50 microns. In Brazil, researchers on MPs follow the global pattern, with most publications focusing on coastal and marine ecosystems [1,36,52,53].

In general, confidence in Brazil's water treatment process is low, as demonstrated by two separate surveys carried out by graduate students studying at the interdisciplinary laboratory as part of their course practical work. The first was a nationwide online survey in 2019 and distributed by social media from April to July 2019. It comprised 1257 participants, mostly belonging to the middle-class, and its results showed that only approximately 4% of the urban population drink unfiltered tap water. The online survey also revealed that most middle-class Brazilian households use domestic water filtration systems, such as activated charcoal filtration (65%), terracotta filter (12%) or purchase bottled drinking water (19%). The second, an in-person survey, was conducted as part of fieldwork performed by graduate students from the University of Brasilia in low-income residential areas on

Brasilia's outskirts. In this group, over 50% of residents drank tap water without any previous treatment, while almost everyone else used terracotta filters. About 80% of people did not use filtered water to wash fruits and vegetables or cook in both groups. One of the main preoccupations is waterborne parasites such as giardia, but boiling water (during the cooking process) kills these parasites (unpublished data). However, as Munno et al. [54] and Avio et al. [55] demonstrated, boiling water does not eliminate MPs. Although not inquired at the time of sample collection, it is safe to assume that juices prepared in commercial establishments are prepared with tap water with no previous filtration in addition to food.

Although our study is not the first to investigate MP contamination in tap water, it is the first analysis in the main Brasilia regions. Therefore, it is not surprising that our results came back positive for MP contamination since previous tap and bottled water contamination studies have also shown MP contamination in bottle and tap water [4,15,18,35,37,38]. What was surprising was that the number of MP particles detected in our samples greatly exceeded the number of MPs found in previous studies. Organic contamination was considered remote, given the precautions set in place, and negative blanks were randomly mixed in while samples were processed. However, even if contamination by organic matter occurred, the number of particles is still significant [49]. To ensure samples were not contaminated, negative blanks were randomly mixed in during sample processing. The next step in the analysis would be identifying MPs by FT-IR method or by Raman spectroscopy. Still, regrettably, none of these techniques are presently available in our laboratory, potentially limiting our study.

Among the several studies on this topic, one of the most extensive was performed by Kosuth et al. [18]. The authors analyzed 159 samples of tap water from 14 countries. Their results showed that 81% of the 159 samples disclosed anthropogenic debris, ranging from 0 to 61, with an overall mean of 5.45 particles/L. Another interesting study was performed by Mason et al. [15]. These authors tested different brands of bottled water from 19 locations in nine other countries. They used Nile Red tagging and a sophisticated software program entitled 'Galaxy Count' to enumerate even the smaller MPs. Their results showed that 93% of the 259 bottles analyzed showed some degree of MP contamination. The densities of MP contamination from 17 bottles ranged from no contamination to one bottle with an excess of 10,000 particles per liter.

We cannot satisfactorily explain the large variability in the number of MP particles found when comparing the findings of the two areas surveyed, since both receive water from the same water treatment plant. We can only speculate that the difference in the collection days and the material of the water tank were a possible influence. The buildings on the South Wing are, in general, older, and their plumbing is mainly made up of galvanized steel pipes. In the North Wing, which had an accelerated development in the 1970s and 1980s, the plumbing used in buildings is mainly PVC.

Finally, it is not the present study's scope to elaborate on the potentially deleterious effects of prolonged ingestion of micro and nanoplastics. MP pollution is a topic of intense research, with numerous studies and reviews at a global level [4,12,13,37]. Evidence from animal studies points out that not all MPs pass inert through the gastrointestinal tract [39] [56–59] and have found evidence of effects to the gut microbiota and inflammation [57,60,61]. A small portion can be absorbed and enter the systemic circulation deposited in organs and tissues [59], and Ragusa et al. found evidence of microplastic in the human placenta [62]. Chronic exposure is also a significant concern due to a cumulative effect [20,63]. An in vivo model demonstrates that MPs are resistant to chemical degradation [64,65]. Furthermore, they can accumulate, resisting mechanical clearance and becoming lodged and embedded [20]. Their biopersistence is an essential factor contributing to human health risk [20,66]. Furthermore, their uptake and toxicity have been studied in mammalian model systems [67]. The finding suggests they can translocate across living cells to the lymphatic and/or circulatory system [68–70], potentially accumulating in secondary organs [71,72] or impacting the immune system [1,20]. Considering the presence

of MPs in the environment and the presence of MPs in drinking water, future studies are necessary to address its consequences for human and animal health.

This study presents some limitations as the method used did not identify the size, shape and composition of MP particles because of the technique used. Moreover, as a preliminary study, we did not collect water from the entire city but only the two main sectors. Due to the distance among the water collection points, water was collected on different days, but in the shortest possible time.

## 5. Conclusions

In conclusion, this study's results reveal a disturbing amount of MP particles in the tap water in an affluent area of Brazil's capital city. Particles were found in 100% of the samples. This surprising number of MP particles in tap water is especially concerning considering that tap water is not the only source of MPs to which people are exposed. It is also concerning because Brasilia is a new and planned city. Further studies are needed in other areas of Brasilia and in different cities in Brazil to determine exposure and the toxicological consequences of these high levels of micro and nanoplastic particles in the general population, especially for susceptible populations, such as children and pregnant women. More studies are needed on the impact of MP concentrations in drinking water on public health.

**Author Contributions:** Conceptualization: C.B.P., M.A.A.L.S.A., M.H.R.T., R.P.Z. Methodology: C.B.P., M.A.A.L.S.A., M.H.R.T., R.P., R.P.Z., G.S.C.P. Validation: R.P.Z., G.S.C.P., M.H.R.T. Formal Analysis: C.B.P., M.A.A.L.S.A., M.H.R.T., R.P., G.S.C.P. Investigation: C.B.P., G.S.C.P., M.A.A.L.S.A. Resources: R.P., L.G., M.H. Supervision: R.P., L.G., M.H., G.S.C.P. Project Administration: C.B.P., R.P., L.G., M.H. All authors have read and agreed to the published version of the manuscript.

**Funding:** This research received no external funding.

**Institutional Review Board Statement:** The study was conducted according to the guidelines of the Declaration of Helsinki, and approved by the Institutional Review Board (or Ethics Committee) of NAME OF INSTITUTE (protocol code 01452010.8.0000.0030 09/23/2017.

**Informed Consent Statement:** Not applicable.

**Data Availability Statement:** The study did not report any data.

**Acknowledgments:** We would like to thank Sherri Masson, one of the leading researchers in plastic pollution, for the inspiration and help.

**Conflicts of Interest:** The authors declare no conflict of interest.

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
