# Peer review of "Presence and Quantification of Microplastic in Urban Tap Water: A Pre-Screening in Brasilia, Brazil"

_sustainability, doi:10.3390/su13116404_

Round 1

Reviewer 1 Report

  1. The sample collected in same area (north wing and south wing) and the potable water come from same water treatment plant. This sample area was not representing the city of Brasilia since the area of sampling was very closed without any diversity of residential and water treatment plant. The sampling location is very crucial to gain the precise result and representative of whole city.
  2. The result showed only the occurrence of microplastic using camera, without any analysis of size, shape or type of polymer.

Author Response

Thank you for taking your time to read our study and comment on it.

  1. The sample collected in same area (north wing and south wing) and the potable water come from same water treatment plant. This sample area was not representing the city of Brasilia since the area of sampling was very closed without any diversity of residential and water treatment plant. The sampling location is very crucial to gain the precise result and representative of whole city.

Yes – that is true. We only sample the main (central) area of Brasilia. However – the results were not uniform. We found a lot more in the north wing.

2. The result showed only the occurrence of microplastic using the camera, without any analysis of size, shape, or type of polymer.

Unfortunately, we do not have the technology necessary to analyze size, shape, or what kind of polymer. Our objective was to analyze the presence and quantify microplastics.

Reviewer 2 Report

Please see my comments in the file attached.

Author Response

Reviewer 2

Thank you for taking your time to read and comment on our study.

  1. Why stress on residential and commercial, what is the difference between them. why not use municipal source?

It is not stressing as much as it is determining the study area.

  1. State the detection limit in terms of particle size.

Unfortunately, we cannot. We lack the technology.

  1. ideas that are unknown and do not fall under the current investigation is not needed in the abstract.

Removed

  1. multiple citations of one idea is not needed unless it is a review paper.

We prefer to over than under refer our work.

  1. He et al is one reference.

Reference number 5 is He

  1. never forever? quantify "never"

We do in the following sentence.

  1. what is the time frame for this amount?

As stated the in the paragraph above the sentence; 1945 to an alarming rate of 335   million tons in 2017

  1. the para repeats what had been mentioned

The objective was to emphasize the difficulty due to its interdisciplinary nature.

  1. a map to show this phenomenon should include areas affected by urbanization and water gathering grounds.

Brasilia became a UNESCO World Heritage site due to its architecture. Therefore, whole there has been urbanization in the surrounding area the city has not changed.

  1. what do you mean by randomly? simple random sampling?

We did not pre-determine a collection algorithm.

  1. All from a single WTP? Yes

  1. Have these types of water tanks been accounted for in your study?

No – It is very complicated to do so. That would add an inordinate amount of variables to the samples.

  1. Please define city-wide analysis. Only the S and N Wing regions were included in the sampling exercise.

  1. Do you mean these people boil water instead of filtering it? Yes – it is not uncommon to boil water to kill pathogens.

Round 2

Reviewer 1 Report

All comment have been addressed

Reviewer 2 Report

Authors have mostly addressed my queries.